# Almost Sure Exponential Stability of Uncertain Stochastic Hopfield Neural Networks Based on Subadditive Measures

**Zhifu Jia** [1],[†] and **Cunlin Li** [2,3],[*],[†]

1   School of Sciences and Arts, Suqian University, Suqian 223800, China; jzflzbx@nuaa.edu.cn
2   Ningxia Key Laboratory of Intelligent Information and Big Data Processing, Governance and Social Management Research Center of Northwest Ethnic Regions, North Minzu University, Yinchuan 750021, China
3   School of Mathematical and Computational Science, Hunan University of Science and Technology, Xiangtan 411201, China
*   Correspondence: 2000014@nmu.edu.cn
†   These authors contributed equally to this work.

**Abstract:** For this paper, we consider the almost sure exponential stability of uncertain stochastic Hopfield neural networks based on subadditive measures. Firstly, we deduce two corollaries, using the Itô–Liu formula. Then, we introduce the concept of almost sure exponential stability for uncertain stochastic Hopfield neural networks. Next, we investigate the almost sure exponential stability of uncertain stochastic Hopfield neural networks, using the Lyapunov method, Liu inequality, the Liu lemma, and exponential martingale inequality. In addition, we prove two sufficient conditions for almost sure exponential stability. Furthermore, we consider stabilization with linear uncertain stochastic perturbation and present some exceptional examples. Finally, our paper provides our conclusion.

**Keywords:** Hopfield neural networks; chance theory; almost sure exponential stability; Lyapunov method

## 1. Introduction

An artificial neural network (ANN) is a computational model inspired by the human brain. ANNs comprise interconnected neurons that process and transmit information. ANNs excel in parallel processing and handling complex, nonlinear problems. ANNs learn from data, recognize patterns, and solve tasks like image recognition and natural language processing. With different architectures such as feedforward, recurrent, and convolutional networks, ANNs have become a crucial component of modern artificial intelligence, enabling machines to learn, adapt, and perform tasks that have traditionally required human intelligence. The Hopfield neural network, as a type of ANN [1], has witnessed steady advancement and intensive investigation over the past few decades, leading to a rich reservoir of research outcomes that have found widespread applications across diverse domains, including combination optimization [2], signal processing [3], pattern recognition [4], and robust control [5]; however, the successful application of neural networks in these fields is closely linked to their dynamic behavior, and stochastic stability is the most important property [6–13]. The above literature shows that the ability of a neural network to maintain stochastic stability (exponential stability and instability [6], exponential stability with time delay [7,8], global stability of stochastic high-order neural networks [9], mean square exponential stability with time-varying delays [10], mean square global asymptotic stability with distributed delays [11], and almost sure exponential stability [12,13]) is crucial for its overall performance, especially when dealing with complex

processes. Hence, significant efforts have been directed towards exploring and enhancing the stability of neural networks.

It is well known that stability is the crucial property of stochastic neural networks, which are often affected simultaneously by parameter uncertainties and random interference factors that can impact their stability due to reasons such as system modeling, measurement errors, and system linearization, as documented in Refs. [14–17]. For example, Huang et al. [14] examined the exponential stability analysis of uncertain stochastic neural networks with multiple delays, and Wang et al. [15] studied the exponential stability of uncertain stochastic neural networks with mixed time delays. Chen et al. [16] investigated the mean square exponential stability of uncertain stochastic delayed neural networks, and Syed [17] surveyed the stochastic stability of uncertain recurrent neural networks with Markovian jumping parameters. However, these studies [14–17] only focused on the robust stability and asymptotic stability of stochastic neural networks with uncertain parameters, while the almost sure exponential stability of neural networks with both uncertain and random disturbances remains unexplored.

As noted above, the stochastic differential equation is a good tool for describing the stability of a stochastic neural network, and the dynamics of the stochastic differential system may be influenced by many other unknown, uncertain, and random disturbances. To address these, Itô [18] established the theory of stochastic analysis and stochastic differential equations with the Wiener process based on additive measures. Over the past 70 years, stochastic differential equations have matured, both in theory and practice, and they have become a vital tool in fields such as physics, systems science, management science, finance, and space science, especially the development of stochastic stability, as in [19–22]. An uncertain process, on the other hand, is a sequence of uncertain variables, with subadditive measures, that change over time. Liu [23] introduced the concept of a Liu process, which is the uncertain version of the Wiener process, in 2008. The Liu process is a Lipschitz continuous process with independent and steady increase properties, and its increments follow an uncertain normal distribution. Based on this process, Liu [24] introduced the chain rule in the process of uncertainty analysis to study the differentials and integrals of uncertain process functions, as well as a class of differential equations driven by standard Liu processes called uncertain differential equations [25]. Consequently, the stability of uncertain differential equations was discussed. When faced with a system that exhibits both uncertainty and randomness simultaneously, the noise should be modeled using the Wiener–Liu process, and the system evolution can be described through a hybrid differential equation, leading to the development of uncertain stochastic hybrid neural network systems [26]. In 2013, Liu [27] first introduced chance theory to investigate such uncertain stochastic systems based on subadditive measures, and subsequent works by Fei et al. [28,29] have further explored the use of the Wiener–Liu process and the Itô–Liu formula in uncertain stochastic differential equations. Researchers have made progress in studying various forms of the stability of stochastic neural networks based on additive measures, but the analysis of indeterminate neural networks, including both random and uncertain factors, requires chance theory's subadditive measures. This paper will review some research results based on chance theory, exploring the stability of uncertain stochastic neural networks using the Itô–Liu formula and the Lyapunov method. The main contributions of this paper are the extension of two corollaries of the Itô–Liu formula under subadditive measures, the introduction of the concept of almost sure exponential stability for uncertain stochastic Hopfield neural networks for the first time, and the consideration of sufficient conditions for almost sure exponential stability and stabilization with linear uncertain stochastic perturbation.

In Section 2, we recall some results about Hopfield neural networks and some concepts, lemmas, theorems, and corollaries about chance theory, which are essential for our analysis. In Section 3, we present our main results about the almost sure exponential stability of uncertain stochastic neural networks. In Section 4, we present our conclusion.

## 2. Preliminaries

### 2.1. The Explanation of Symbols

We add the table of momenclature so that we could relate to symbols used in the paper easily (Table 1).

**Table 1.** The explanation of symbols related to this paper.

| Numbers | The Symbols of This Paper | The Explanation of Symbols |
|:---:|:---:|:---:|
| 1 | $u_i(k)$ | voltage on the input of the ith neuron |
| 2 | $F_i$ | input capacitance |
| 3 | $T_{ij}$ | connection matrix element |
| 4 | $f_i(u)$ | nondecreasing transfer function |
| 5 | $\varsigma_i$ | slope of $f_i(u)$ at $u = 0$ |
| 6 | $\mathcal{M}$ | uncertain measure |
| 7 | $k$ | time |
| 8 | $C_k$ | Liu process |
| 9 | $Ch$ | chance measure |
| 10 | $\mathcal{P}$ | probability measure |
| 11 | $W_k$ | Wiener process |
| 12 | $Z_k$ | uncertain process or uncertain stochastic process |
| 13 | sup | supremum |

### 2.2. The Basic Knowledge

A Hopfield neural network [1] can be described in the form of an ordinary differential equation as follows:

$$F_i \dot{u}_i(k) = -\frac{1}{R_i} u_i(k) + \sum_{j=1}^{m} T_{ij} f_j(u_j(k)), \ 1 \le i \le m, \ k \ge 0, \tag{1}$$

where $u_i(k)$ denotes the voltage on the input of the ith neuron, $F_i$ denotes the input capacitance, $T_{ij}$ is the connection matrix element, $f_i(u)$ is a nondecreasing transfer function, see Table 1, and $f_i(0) = 0$; the following $\varsigma_i$ is the slope of $f_i(u)$ at $u = 0$, satisfying

$$u f_i(u) \ge 0, \ |f_i(u)| \le 1 \wedge \varsigma_i |u|, \ -\infty < u < +\infty. \tag{2}$$

where $1 \wedge \varsigma_i |u|$ determines the upper bound of the function $|f_i(u)|$ and is denoted by

$$e_i = \frac{1}{F_i R_i}, \ b_{ij} = \frac{T_{ij}}{F_i},$$

then,

$$\dot{u}_k = -E u_k + B f(u_k), \ k \ge 0, \tag{3}$$

where

$$u_k = (u_{1k}, \cdots, u_{mk})^T, \ E = diag.(e_1, \cdots, e_m), \ B = (b_{ij})_{m \times m}, \ f(u) = (f_1(u_1), \cdots, f_m(u_m))^T.$$

Furthermore,

$$e_i = \sum_{j=1}^{m} |b_{ij}|, \ 1 \le i \le m. \tag{4}$$

Itis easy to know that for any given initial case $u_0 = z_0 \in R^m$, the equation has a unique solution. In particular, the equation is unique equilibrium solution $u_0 = 0$. In other

words, the zero point is the equilibrium point of the neural network system. The aim of this paper is to investigate the uncertain stochastic effects on the stability. The following reviews chance theory including some concepts, lemmas, theorems, and corollaries which are essential for our analysis.

Let $\Gamma$ be a nonempty set, and $\mathcal{L}$ a $\sigma$-algebra over $\Gamma$. Each element $\Lambda$ in $\mathcal{L}$ is called an event and $\mathcal{M}\{\Lambda\}$ is the belief degree. The uncertain measure dealing with belief degree satisfies the following axioms [23,25]:

Axiom 1 (Normality Axiom). $\mathcal{M}\{\Lambda\} = 1$ for the universal set $\Gamma$.

Axiom 2 (Duality Axiom). $\mathcal{M}\{\Lambda\} + \mathcal{M}\{\Lambda^c\} = 1$ for any event $\Lambda$.

Axiom 3 (Subadditivity Axiom). For every countable sequence of events $\Lambda_1, \Lambda_2, \cdots$,

$$\mathcal{M}\left\{\bigcup_{i=1}^{\infty} \Lambda_i\right\} \leq \sum_{i=1}^{\infty} \mathcal{M}_k\{\Lambda_i\}$$

holds.

Axiom 4 (Product Axiom). Let $(\Gamma_j, \mathcal{L}_j, \mathcal{M}_j)$ be uncertainty spaces for $j = 1, 2, \cdots$. The product uncertain measure $\mathcal{M}$ is an uncertain measure satisfying

$$\mathcal{M}\left\{\prod_{j=1}^{\infty} \Lambda_j\right\} = \bigwedge_{j=1}^{\infty} \mathcal{M}_j\{\Lambda_j\}.$$

where $\Lambda_j$ are arbitrary events chosen from $\mathcal{L}_j$ for $j = 1, 2, \cdots$, respectively.

**Remark 1.** *Axioms 1 and 2 are similar to probability theory, and axioms 3 and 4 are fundamentally different from probability theory. In particular, axiom 3 embodies subadditivity, which is different from the additivity of probability theory, and the product axiom of axiom 4 embodies the minimization operation, which is different from the product axiom of probability theory. The detailed analysis can be found in Refs. [23,25].*

**Definition 1** ([23]). *An uncertain variable is a measurable function $\xi$ from an uncertainty space $(\Gamma, \mathcal{L}, \mathcal{M})$ to the set of real numbers, i.e., for any Borel set $B$ of real numbers, the set*

$$\{\xi \in B\} = \{\gamma \in \Gamma | \xi(\gamma) \in B\}$$

*is an event.*

**Definition 2** ([23]). *Let $T$ be an index set and $(\Gamma, \mathcal{L}, \mathcal{M})$ an uncertainty space. An uncertain process is a measurable function from $T \times (\Gamma, \mathcal{L}, \mathcal{M})$ to the set of real numbers such that $\{Z_k \in B\}$ is an event for any Borel set $B$ for each time $k$.*

**Definition 3** ([23]). *An uncertain process $C_k$ is said to be a Liu process if*
*(i) $C_0 = 0$ and almost all sample paths are Lipschitz continuous;*
*(ii) $C_k$ has stationary and independent increments;*
*(iii) every increment $C_{r+k} - C_r$ is a normal uncertain variable with expected value 0 and variance $k^2$, whose uncertainty distribution is*

$$\Phi(x) = \left(1 + \exp\left(\frac{-\pi x}{\sqrt{3}k}\right)\right)^{-1}, \ x \in \ R.$$

**Definition 4** ([23]). *Let $Z_k$ be an uncertain process with respect to time $k$ and $C_k$ be a Liu process with respect to time $k$. For any partition of closed interval $[a, b]$ with $a = k_1 < k_2 < \cdots < k_{j+1} = b$, the mesh is written as*

$$\Delta = \max_{1 \leq i \leq j} |k_{i+1} - k_i|.$$

*Then, the uncertain integral of $Z_k$ with respect to $C_k$ is*

$$\int_a^b Z_k dC_k = \lim_{\Delta \to 0} \sum_{i=1}^{j} Z_{k_i} \cdot (C_{k_{i+1}} - C_{k_i})$$

*provided that the limit exists almost surely and is finite. In this case, the uncertain process $Z_k$ is said to be integrable.*

**Lemma 1** ([25] (Liu inequality)). *Let $C_k$ be a Liu process on uncertainty space $(\Gamma, \mathcal{L}, \mathcal{M})$. Then, there exists an uncertain variable $K$ such that $K(\gamma)$ is a Lipschitz constant of the sample path $C_k(\gamma)$ for each $\gamma$,*

$$\lim_{x \to +\infty} M\{\gamma \in \Gamma | K(\gamma) \leq x\} = 1$$

*and*

$$M\{\gamma \in \Gamma | K(\gamma) \leq x\} \geq 2\Phi(x) - 1.$$

**Lemma 2** ([26] (Liu lemma)). *Suppose that $C_k$ is a Liu process, and $Z_k$ is an integrable uncertain process on $[a, b]$ with respect to $k$. Then, the inequality*

$$|\int_a^b Z_k(\gamma) dC_k| \leq K(\gamma) \int_a^b |Z_k(\gamma)| dk$$

*holds, where $K(\gamma)$ is the Lipschitz constant of the sample path $Z_k(\gamma)$.*

Let $(\Omega, \mathcal{F}, P)$ be a complete probability space with a filtration $\{\mathcal{F}_k\}_{k \in [0,T]}$ satisfying the usual conditions, that is, it is increasing and right continuous while $\mathcal{F}_0$ contains all $\mathcal{P}$-null sets.

Let $(\Gamma, \mathcal{L}, \mathcal{M})$ be an uncertainty space where normality, duality, subadditivity, and product measure axioms are given. Let $C_k$ be Liu Liu process defined on $(\Gamma, \mathcal{L}, \mathcal{M})$. The Liu process filtration $\{\mathcal{L}_k\}_{k \in [0,T]}$ is the sub-$\sigma$-field family $(\mathcal{L}_k, k \in [0,T])$ of $\mathcal{L}$ satisfying the usual conditions. It is generalized by $\sigma(C_s : s \leq k)$ and $\mathcal{M}$-null sets of $\mathcal{L}$, $\mathcal{L}_T = \mathcal{L}$.

Liu [27] first introduced chance theory to investigate a hybrid system with both uncertainty about belief degree and randomness. To investigate the uncertain stochastic differential systems, Fei [29] extended a filtered chance space $(\Gamma \times \Omega, \mathcal{L} \otimes \mathcal{F}, (\mathcal{L}_k \otimes \mathcal{F}_k)_{k \in [0,T]}, \mathcal{M} \times \mathcal{P})$ on which some concepts, theorems, are presented as follows.

**Definition 5** ([29]). *(i) Let B be a Borel set; an uncertain random variable is a measurable function $\xi \in R^p$ (or $R^{p \times m}$) from a chance space*

$$(\Gamma \times \Omega, \mathcal{L} \otimes \mathcal{F}, \mathcal{M} \times P)$$

*to $R^p$ (or $R^{p \times m}$), that is, $\forall B \in R^p$ (or $R^{p \times m}$), so the set*

$$\{\xi \in B\} = \{(\gamma, \omega) \in \Gamma \times \Omega : \xi(\gamma, \omega) \in B\} \in \mathcal{L} \otimes \mathcal{F}.$$

*(ii) $\forall B$, $\{\xi \in B\}$ is an uncertain random event with chance measure*

$$\mathrm{Ch}\{\xi \in B\} = \int_0^1 \mathcal{P}\{\omega \in \Omega \,|\, \mathcal{M}\{\gamma \in \Gamma \,|\, \xi(\gamma, \omega) \in B\} \geq x\} dx.$$

**Definition 6** ([29]). *(a) An uncertain stochastic process is essentially a sequence of uncertain variables indexed by time. For each time $k \in [0, T]$, if $Z_k$ is an uncertain random variable, then we call $Z_k$ an uncertain stochastic process (or hybrid process). If the sample paths of $Z_k$ are continuous functions of $k$ for almost all $(\gamma, \omega) \in \Gamma \times \Omega$, then we call it continuous.*

*(b) If $Z(k, \gamma)$ is $\mathcal{F}_k$-measurable for all $k \in [0, T]$, $\gamma \in \Gamma$, then we call it $\mathcal{F}_k$-adapted. Further, if $Z(k)$ is $\mathcal{L}_k \otimes \mathcal{F}_k$-measurable for all $k \in [0, T]$, then we call it $\mathcal{L}_k \otimes \mathcal{F}_k$-adapted (or adapted).*

*(c) If the uncertain stochastic process is measurable related to the σ-algebra*

$$\Im(\mathcal{L}_k \otimes \mathcal{F}_k)$$
$$= \{A \in B([0,T]) \otimes \mathcal{L} \otimes \mathcal{F} : A \cap ([0,k] \times \Gamma \times \Omega) \in B([0,k]) \otimes \mathcal{L}_k \otimes \mathcal{F}_k\}.$$

*then we call it progressively measurable.*

*Further, if the uncertain stochastic process $Z(k) : \Gamma \times \Omega \to R^p$ (or $Z(k) : \Gamma \times \Omega \to R^{p \times m}$ is progressively measurable and satisfies $\forall k \in [0,T]$, $E[\int_0^T |Z_k|^2 dk]$, then we call it $L^2$-progressively measurable, where $L^2(0,T;R^p)$ (or $L^2(0,T;R^{p \times m})$) denotes the set of $L^2$-progressively measurable uncertain random processes.*

**Definition 7** ([28])**.** *Let $W_k$ be a Wiener process and $C_k$ a Liu process. Then, $\mathcal{H}_k = (W_k, C_k)$ is called a Wiener–Liu process. The Wiener–Liu process is said to be standard if both $W_k$ and $C_k$ are standard.*

**Definition 8** ([28])**.** *Let $Z_k = (\hat{Z}_k, \tilde{Z}_k)$, where $\hat{Z}_k$ and $\tilde{Z}_k$ are scalar uncertain stochastic processes, and let $\mathcal{H}_k = (W_k, C_k)$ be a standard Wiener–Liu process. For any partition of a closed interval $[a,b]$ with $a = k_1 < k_2 < \cdots < k_{N+1} = b$, the mesh is written as*

$$\Delta = \max_{1 \le i \le N} |k_{i+1} - k_i|.$$

*Then, the uncertain stochastic integral of $Z_k$ with respect to $\mathcal{H}_k$ is*

$$\int_a^b Z_k d\mathcal{H}_k = \lim_{\Delta \to 0} \sum_{i=1}^N (\hat{Z}_{k_i} \cdot (W_{k_{i+1}} - W_{k_i}) + \tilde{Z}_{k_i} \cdot (C_{k_{i+1}} - C_{k_i}))$$

*provided that the limit exists almost surely and is finite. In this case, the uncertain stochastic process $Z_k$ is said to be integrable.*

**Remark 2.** *The uncertain stochastic integral may also be written as follows:*

$$\int_a^b Z_k d\mathcal{H}_k = \int_a^b (\hat{Z}_k dW_k + \tilde{Z}_k dC_k). \tag{5}$$

The following theorem results in the Itô–Liu formula of the one-dimensional case.

**Theorem 1** ([28] (Itô–Liu formula))**.** *Let $\mathcal{H}_k$ be a Wiener–Liu process given by*

$$\mathcal{H}_k = (Z_k, \bar{Z}_k) = (\mu_1 k + \sigma_1 W_k, \mu_2 k + \sigma_2 C_k).$$

*Let $W_k$ be a Wiener process and $C_k$ a Liu process, and $g(k, z, \bar{z})$ a twice continuously differentiable function. Define $G_k = g(k, Z_k, \bar{Z}_k)$. Then, we have the following chain rule:*

$$dG_k = \frac{\partial g}{\partial k}(k, Z_k, \bar{Z}_k)dk + \frac{\partial g}{\partial z}(k, Z_k, \bar{Z}_k)dW_k + \frac{\partial g}{\partial \bar{z}}(k, Z_k, \bar{Z}_k)dC_k$$
$$+ \frac{1}{2}\frac{\partial^2 g}{\partial z^2}(k, Z_k, \bar{Z}_k)dk.$$

Using Theorem 1, we can easily obtain the following two corollaries.

**Corollary 1.** *The infinitesimal increments $dW_k$ and $dC_k$ may be replaced with the derived Wiener–Liu process,*

$$Z_k = \int_0^k \mu_u du + \int_0^k \alpha_u dW_u + \int_0^k \beta_u dC_u,$$

where $\mu_k$ and $\beta_k$ are absolutely integrable uncertain stochastic processes, and $\alpha_k$ is a square integrable uncertain stochastic process; then, $\forall \Phi \in \mathcal{C}^2(\Re)$ ($\mathcal{C}^2$ means second-order continuous differentiable), thus producing

$$\Phi(Z_k) = \Phi(Z_0) + \int_0^k \Phi'(Z_u)\mu_u du + \int_0^k \Phi'(Z_u)\alpha_u dW_u$$
$$+ \int_0^k \Phi'(Z_u)\beta_u dC_u + \frac{1}{2}\int_0^k \Phi''(Z_u)\alpha_u^2 du.$$

Let $W_k = (W_{1k}, W_{2k}, \cdots, W_{pk})$ and $C_k = (C_{1k}, C_{2k}, \cdots, C_{qk})$ be a $p$-dimensional standard Wiener process and a $q$-dimensional standard Liu process, respectively. If $r_i$ and $v_{ij}$ are absolute integrable hybrid processes, and $w_{ij}$ are square integrable hybrid processes, for $i = 1, 2, \cdots, m, j = 1, 2, \cdots, q$, then the $m$-dimensional hybrid process $\mathbf{Z}_k = (Z_{1k}, Z_{2k}, \cdots, Z_{mk})$ is given by

$$\begin{cases} dZ_{1k} = r_1 dk + \sum\limits_{j=1}^{p} w_{1j} dW_{jk} + \sum\limits_{j=1}^{q} v_{1j} dC_{jk} \\ \vdots \qquad \vdots \qquad\qquad\qquad \vdots \\ dZ_{mk} = r_m dk + \sum\limits_{j=1}^{p} w_{mj} dW_{jk} + \sum\limits_{j=1}^{q} v_{mj} dC_{jk}, \end{cases}$$

or, in matrix notation, simply

$$d\mathbf{Z}_k = \mathbf{r}dk + \mathbf{w}d\mathbf{W}_k + \mathbf{v}d\mathbf{C}_k,$$

where

$$\mathbf{r} = \begin{pmatrix} r_1 \\ \vdots \\ r_m \end{pmatrix}, \mathbf{w} = \begin{pmatrix} w_{11} \cdots w_{1p} \\ \vdots \quad \vdots \\ w_{m1} \cdots w_{mp} \end{pmatrix}, \mathbf{v} = \begin{pmatrix} v_{11} \cdots v_{1q} \\ \vdots \quad \vdots \\ v_{m1} \cdots v_{mq} \end{pmatrix}, d\mathbf{W}_k = \begin{pmatrix} dW_{1k} \\ \vdots \\ dW_{pk} \end{pmatrix}, d\mathbf{C}_k = \begin{pmatrix} dC_{1k} \\ \vdots \\ dC_{qk} \end{pmatrix}.$$

**Corollary 2.** *Assume m-dimensional hybrid process $\mathbf{Z}_k$ is given by*

$$d\mathbf{Z}_k = \mathbf{r}dk + \mathbf{w}d\mathbf{W}_k + \mathbf{v}d\mathbf{C}_k,$$

*Let $g(k, z_1, \cdots, z_m)$ be a multivariate continuously differentiable function. Define $G_k = g(k, Z_{1k}, \cdots, Z_{mk})$. Then,*

$$dG_k = \frac{\partial g}{\partial k}(k, Z_{1k}, \cdots, Z_{mk})dk + \sum_{i=1}^{m} \frac{\partial g}{\partial z_i}(k, Z_{1k}, \cdots, Z_{mk})dZ_{ik}$$
$$+ \frac{1}{2}\sum_{i=1}^{m}\sum_{j=1}^{m} \frac{\partial^2 g}{\partial z_i \partial z_j}(k, Z_{1k}, \cdots, Z_{mk})dZ_{ik}dZ_{jk},$$

*where $dW_{ik}dW_{jk} = \delta_{ij}dk, dW_{ik}dk = dkdW_{ik} = dC_{\imath k}dC_{\jmath k} = dkdC_{\imath k} = dW_{ik}dC_{\imath k} = 0$, for $i, j = 1, 2, \cdots, p, \imath, \jmath = 1, 2, \cdots, q$. And*

$$\delta_{ij} = \begin{cases} 0, i \neq j \\ 1, i = j \end{cases} \tag{6}$$

In other words, it can be expressed as

$$dG_k = \frac{\partial g}{\partial k}(k, Z_{1k}, \cdots, Z_{mk})dk + \sum_{i=1}^{p} \frac{\partial g}{\partial z_i}(k, W_{1k}, \cdots, W_{pk}, C_{1k}, \cdots, C_{qk})dW_{ik}$$

$$+ \sum_{j=1}^{q} \frac{\partial g}{\partial z_{m+j}}(k, W_{1k}, \cdots, W_{pk}, C_{1k}, \cdots, C_{qk})dC_{jk}$$

$$+ \frac{1}{2} \sum_{i=1}^{p} \frac{\partial^2 g}{\partial z_i^2}(k, W_{1k}, \cdots, W_{pk}, C_{1k}, \cdots, C_{qk})dk.$$

**Definition 9** ([28]). *Suppose $W_k$ is a standard , $C_k$ is a standard process, and $f$, $g$, and $h$ are some given functions. Then,*

$$dZ_k = f(k, Z_k)dk + g(k, Z_k)dW_k + h(k, Z_k)dC_k \tag{7}$$

*is called an uncertain stochastic differential equation.*

## 3. Main Results

Let us consider a hypothetical scenario in which an uncertain stochastic perturbation is introduced to the neural network, and as a result, the perturbed network can be modeled using an uncertain stochastic differential equation.

$$\begin{cases} dz(k) = [-Ez(k) + Bf(z(k))]dk + g(z(k))dW(k) + h(z(k))dC(k), k \geq 0, \\ z(0) = z_0 \in R^m, \end{cases} \tag{8}$$

where $W(k) = (W_1(k), \ldots, W_n(k))^T$ denotes an $n$-dimensional Wiener process and $f : R^m \to R^{m \times n}$ (i.e. $f(z) = (f_{ij}(z))_{m \times n}$. Additionally, let $C(k) = (C_1(k), \ldots, C_n(k))^T$ and $h : R^m \to R^{m \times n}$ i.e., $h(z) = (h_{ij}(z))_{m \times n}$. In addition, $g(z)$ and $h(z)$ satisfy the Lipschitz continuous and satisfy the linear growth condition. Consequently, we can deduce from Refs. [28,29] that for $k \geq 0$, Equation (8) possesses a unique global solution $z(k, z_0)$, assuming $g(0) = h(0) = 0$ for the sake of stability in this paper. As a result, Equation (8) possesses an equilibrium solution $z(k, 0) = 0$. Additionally, when $z_0 \neq 0$, the uniqueness exists with chance measure one, that is, $z(k, z_0) \neq 0$ for all $k \geq 0$ almost surely.

In contrast to Equation (3), Equation (8) represents a system with an uncertain stochastic perturbation. It is intriguing to explore the influence of uncertain stochastic perturbation on the stability characteristics of the neural network. In the next section, we will delve into these issues in great depth.

### 3.1. Almost Sure Exponential Stability

**Definition 10.** *Firstly, we assume that Equation (8) has a solution $z_0 = 0$. Further, we assume that there exist two measure sets, $\mathcal{M}\{\Gamma_{\epsilon_1}\}$ and $\mathcal{P}\{\Omega_{\epsilon_2}\}$, such that for any $\epsilon_1, \epsilon_2 > 0$ and for all $\forall \gamma \in \Gamma \setminus \Gamma_{\epsilon_1}$ and $\forall \omega \in \Omega \setminus \Omega_{\epsilon_2}$, the nonzero solution $z(k, z_0)$ of Equation (8) when $z_0 \neq 0$ satisfies the following condition:*

$$\limsup_{k \to \infty} \frac{1}{k} \ln(|z(k, z_0)|) < 0, \tag{9}$$

*then, we call the uncertain stochastic neural network (8) almost surely exponentially stable, simply denoted as*

$$\limsup_{k \to \infty} \frac{1}{k} \ln(|z(k, z_0)|) < 0, a.s. \tag{10}$$

**Theorem 2.** *Assume there exists a symmetric positive definite matrix $P = (p_{ij})_{m \times m}$ and some constants $\mu \in R$ and $\rho_1, \rho_2, H > 0$ such that*

$$2z^T P[-Ez + Bf(z)] + tr[g^T(z)Pg(z)] \leq \mu z^T Pz, \tag{11}$$

$$z^T P g(x) g^T(z) P z \geq \rho_1 (z^T P z)^2,$$ (12)

$$|z^T P h(x)| \leq \frac{\rho_2}{n} z^T P z$$ (13)

*for all $z \in R^m$. Then, the solution of Equation (8) satisfies*

$$\limsup_{k \to \infty} \frac{1}{k} ln(|z(k, z_0)|) \leq -(\rho_1 - H\rho_2 - \frac{\mu}{2}) \; a.s.$$ (14)

*whenever $z_0 \neq 0$. Especially, if $\rho_1 - H\rho_2 > \mu/2$, then the stochastic neural network (8) is almost surely exponentially stable.*

**Proof.** Take the Lyapunov function

$$V(z, k) = z^T P z.$$

Choose any nonzero value of $z_0$ and define $z(k, z_0)$ as $z_k$. It follows from the fact that there is only one possible solution that $z(k)$ will almost surely be nonzero for all $k > 0$. The Itô–Liu formula implies that

$$
\begin{aligned}
&d(\ln[z_k^T P z_k]) \\
&= \frac{1}{z_k^T P z_k} (2z_k^T P[-E z_k + B f(z_k)] + tr[g^T(z_k) P g(z_k)]) dk \\
&\quad - \frac{2}{[z_k^T P z_k]^2} (z_k^T P g(z_k) g^T(z_k) P z_k) dk \\
&\quad + \frac{2}{[z_k^T P z_k]} z_k^T P g(z_k) dW_k \\
&\quad + \frac{2}{[z_k^T P z_k]} z_k^T P h(z_k) dC_k.
\end{aligned}
$$

Considering condition (11), we obtain

$$\ln[z_k^T P z_k] \leq \ln[z_0^T P z_0] + \mu k - 2\langle M_k \rangle + 2M_k + 2N_k, \; a.s.$$ (15)

where

$$N_k = \int_0^k \frac{1}{[z_s^T P z_s]} z_s^T P h(z_s) dC_s$$

for all $k > 0$, where $N_k$ is an uncertain process and $N_0 = 0$, and

$$M_k = \int_0^k \frac{1}{[z_s^T P z_s]} z_s^T P g(z_s) dW_s,$$

which is a continuous martingale that disappears when $k = 0$. This martingale's quadratic variation is denoted by $\langle M_k \rangle$. That is,

$$\langle M_k \rangle = \int_0^k \frac{1}{[z_s^T P z_s]^2} (z_s^T P g(z_s) g^T(z_s) P z_s) ds.$$

By condition (12), we obtain

$$\langle M_k \rangle \geq \rho_1 k.$$ (16)

Let $l = 1, 2 \cdots$ . and $\epsilon \in (0, l)$ be arbitrary. The exponential martingale inequality implies

$$\mathcal{P}(\omega : \sup_{0 \leq k \leq l} [M_k - \epsilon \langle M_k \rangle] > \frac{1}{2\epsilon} \ln l) \leq \frac{1}{l}.$$

Therefore, according to the Borel–Cantelli lemma, it follows that there exists a random integer $l_0(\omega)$ for almost every $\omega \in \Omega$, such that for all $l \geq l_0$, the following holds:

$$\sup_{0 \leq k \leq l} [M_k - \epsilon \langle M_k \rangle] \leq \frac{1}{2\epsilon} \ln l,$$

that is,

$$M_k \leq \epsilon \langle M_k \rangle + \frac{1}{2\epsilon} \ln l, 0 \leq k \leq l.$$

By condition (13), for any event $\gamma \in \Gamma$, we have

$$\begin{aligned}
N_k(\gamma) \leq |N_k(\gamma)| &\leq n \cdot K(\gamma) \int_0^k \frac{1}{[z_s^T P z_s]} z_s^T P h(z_s) ds \\
&\leq n \cdot K(\gamma) \frac{\rho_2}{n} k \\
&= K(\gamma) \rho_2 k,
\end{aligned}$$

where $K(\gamma) = \max\limits_i K_i(\gamma)$, $K_i(\gamma)$ is a Lipschitz constant of $C_{ik}$. By Lemma 1, for $\forall \epsilon > 0$, there exists positive $H = H(\gamma)$, such that

$$\mathcal{M}\{\gamma \in \Gamma | K(\gamma) \leq H\} > 1 - \epsilon,$$

namely, $\forall \epsilon > 0, \exists \Gamma_\epsilon$, such that

$$\gamma \in \Gamma \setminus \Gamma_\epsilon, \ N_k(\gamma) \leq H \rho_2 k.$$

Substituting this into (15) yields

$$\ln[z_k^T P z_k] \leq \ln[z_0^T P z_0] + \mu k - (2 - \epsilon)\langle M_k \rangle - 2H\rho_2 k + \frac{1}{\epsilon} \ln l$$

for all $0 \leq k \leq l$ and $l \geq l_0$, almost surely. By (16), we can obtain that

$$\ln[z_k^T P z_k] \leq \ln[z_0^T P z_0] + \mu k - (2 - \epsilon)\rho_1 k - 2H\rho_2 k + \frac{1}{\epsilon} \ln l$$

for all $0 \leq k \leq l$ and $l \geq l_0$, almost surely. So, for almost all $\omega \in \Omega$, $\gamma \in \Gamma$ if $l - 1 \leq k \leq l$ and $l \geq l_0$, then

$$\frac{1}{k} \ln[z_k^T P z_k] \leq -[(2 - \epsilon)\rho_1 - 2H\rho_2 - \mu] + \frac{1}{l-1}(\ln[z_0^T P z_0] + \frac{1}{\epsilon} \ln l).$$

Letting $\epsilon \to 0$, we obtain

$$\limsup_{k \to \infty} \frac{1}{k} \ln[z_k^T P z_k] \leq -[2\rho_1 - 2H\rho_2 - \mu].$$

Because $P$ is a symmetric positive definite matrix, the minimum eigenvalue $\lambda_{min} > 0$, and then

$$\lambda_{min} |z|^2 \leq z^T P z, \ z \in R^m.$$

Thus

$$\limsup_{k\to\infty} \frac{1}{k} \ln[z_k^T P z_k] \geq \limsup_{k\to\infty} \frac{1}{k} \ln(\lambda_{min}|z_k|^2)$$

$$= \limsup_{k\to\infty} \frac{1}{k}(\ln \lambda_{min} + 2\ln|z_k|)$$

$$= 2\limsup_{k\to\infty} \frac{1}{k} \ln|z_k|.$$

Thus

$$\limsup_{k\to\infty} \frac{1}{k} \ln(|z_k|) \leq -(\rho_1 - H\rho_2 - \frac{\mu}{2}).$$

We complete the proof. □

By Theorem 2, the following two sufficient conclusions can be obtained.

**Theorem 3.** *Suppose* (2) *is satisfied, and there exists a diagonal matrix* $P = diag(p_1, p_2, \cdots, p_m)$ *where* $p_i > 0$ *for all* $i$. *Let* $\mu > 0$, $\rho_1, \rho_2$ *be real numbers, and let the constant* $H > 0$ *such that*

$$tr[g^T(z)Pg(z)] \leq \mu z^T Pz,$$

$$z^T Pg(z)g^T(z)Pz \geq \rho_1(z^T Pz)^2,$$

$$|z^T Ph(z)| \leq \frac{\rho_2}{n} z^T Pz$$

*for all* $z \in R^m$. *Denote by* $\lambda_{max}(Q)$ *the largest eigenvalue of the symmetric matrix* $Q = (q_{ij})_{m\times m}$, *where* $q_{ij}$ *is defined as follows:*

$$q_{ij} = \begin{cases} 2p_i[-e_i + (0 \vee b_{ii})\varsigma_i], & for\ i = j, \\ p_i|b_{ij}|\varsigma_j + p_j|b_{ji}|\varsigma_i, & for\ i \neq j. \end{cases} \tag{17}$$

*Then, the solution of Equation* (8) *satisfies*
*(i) if* $\lambda_{max}(Q) \geq 0$

$$\limsup_{k\to\infty} \frac{1}{k} \ln(|z(k, z_0)|) \leq (\frac{1}{2}[\mu + \frac{\lambda_{max}(Q)}{\min_{1\leq i\leq m} p_i}] + H\rho_2 - \rho_1), \ a.s. \tag{18}$$

*(ii) if* $\lambda_{max}(Q) < 0$

$$\limsup_{k\to\infty} \frac{1}{k} \ln(|z(k, z_0)|) \leq (\frac{1}{2}[\mu + \frac{\lambda_{max}(Q)}{\min_{1\leq i\leq m} p_i}] + H\rho_2 - \rho_1), \ a.s. \tag{19}$$

*whenever* $z_0 \neq 0$.

**Proof.** It holds from (2) that

$$2z^T PAf(x) = 2\sum_{i,j=1}^m z_i p_i b_{ij} f_j(z_j)$$

$$\leq 2\sum_i p_i(0 \vee b_{ii})z_i f_i(z_i) + 2\sum_{i\neq j}|z_i|p_i|b_{ij}|\varsigma_j|z_j|$$

$$\leq 2\sum_i p_i(0 \vee b_{ii})\varsigma_i z_i^2 + \sum_{i \neq j} |z_i|(p_i|b_{ij}|\varsigma_j + p_j|b_{ji}|\varsigma_i)|z_j|.$$

Thus, when $\lambda_{max}(Q) \geq 0$,

$$2z^T P[-Ez + Bf(z)] \leq (|z_1|, \cdots, |z_m|)Q(|z_1|, \cdots, |z_m|)^T$$
$$\leq \lambda_{max}(Q)|z|^2 \leq \frac{\lambda_{max}(Q)}{\min\limits_{1 \leq i \leq m} p_i} z^T Pz.$$

We can easily arrive at conclusion (18) by applying Theorem 2. Additionally, when $\lambda_{max}(Q) < 0$,

$$2z^T P[-Ez + Bf(z)] \leq (|z_1|, \cdots, |z_m|)Q(|z_1|, \cdots, |z_m|)^T$$
$$\leq \lambda_{max}(Q)|z|^2 \leq \frac{\lambda_{max}(Q)}{\min\limits_{1 \leq i \leq m} p_i} z^T Pz.$$

By utilizing Theorem 2 once more, we can arrive at conclusion (19). Hence, we complete the proof. $\square$

**Theorem 4.** *Suppose both* (2) *and* (4) *are satisfied, where $\delta_{ij}$ is defined the same as* (6). *Additionally, assume that there exist m positive numbers $p_1, p_2, \cdots, p_m$ such that*

$$\varsigma_j^2 \sum_{i=1}^m p_i[0 \vee sign(b_{ii})]^{\delta_{ij}}|b_{ij}| \leq p_j e_j, \ 1 \leq j \leq m,$$

*and*

$$tr[g^T(z)Pg(z)] \leq \mu z^T Pz,$$

$$z^T Pg(z)g^T(z) \geq \rho_1(z^T Pz)^2,$$

$$|z^T Ph(z)| \leq \frac{\rho_2}{n} z^T Pz,$$

*where $P = diag.(p_1, p_2, \cdots, p_m)$ and the real numbers $\mu > 0, \rho_1, \rho_2, H > 0$. Then for all $z \in R^m$, the solution of Equation* (8) *satisfies*

$$\limsup_{k \to \infty} \frac{1}{k} ln(|z(k, z_0)|) \leq -(\rho_1 - H\rho_2 - \frac{\mu}{2}) \ a.s. \tag{20}$$

**Proof.** By condition, we can obtain that

$$2z^T PAf(x) = 2\sum_{i,j=1}^m z_i p_i b_{ij} f_j(z_j)$$
$$\leq 2\sum_{i,j=1}^m |z_i|p_i[0 \vee sign(b_{ii})]^{\delta_{ij}}|b_{ij}|\varsigma_j|z_j|$$
$$\leq \sum_{i,j=1}^m p_i[0 \vee sign(b_{ii})]^{\delta_{ij}}|b_{ij}|(z_i^2 + \varsigma_j^2 z_j^2)$$
$$\leq \sum_{i=1}^m p_i(\sum_{j=1}^m |b_{ij}|)z_i^2 + \sum_{j=1}^m (\varsigma_j^2 \sum_{i=1}^m p_j[0 \vee sign(b_{ii})]^{\delta_{ij}}|b_{ij}|)z_j^2$$

$$\leq \sum_{i=1}^{m} p_i e_i z_i^2 + \sum_{j=1}^{m} p_j e_j z_j^2 = 2z^T P E z.$$

Hence

$$2z^T P[-Ez + Bf(z)] + tr[g^T(z)Pg(z)] \leq \mu z^T P z. \tag{21}$$

So, by Theorem 2 again, we complete the proof. $\square$

**Theorem 5.** *Suppose both* (2) *and* (4) *are satisfied. We assume that the network is symmetric, meaning that*

$$|b_{ij}| = |b_{ji}|, \forall 1 \leq i, j \leq m.$$

*Moreover, assume*

$$tr[g^T(z)Pg(z)] \leq \mu|z|^2,$$

$$z^T P g(z) g^T(z) \geq \rho_1 |z|^4,$$

$$|z^T h(z)| \leq \rho_2 |z|^2$$

*hold for all $z \in R^m$, where $\mu > 0$ and $\rho_1, \rho_2, H > 0$ are constants. Then, the solution to Equation* (8) *holds that*

$$\limsup_{k \to \infty} \frac{1}{k} ln(|z(k, z_0)|) \leq -(\rho_1 - H\rho_2 + \hat{e}(1 - \check{\zeta}) - \frac{\mu}{2}) \ a.s. \tag{22}$$

$1 \geq \check{\zeta}$, *or*

$$\limsup_{k \to \infty} \frac{1}{k} ln(|z(k, z_0)|) \leq -(\rho_1 - H\rho_2 - \check{e}(\check{\zeta} - 1) - \frac{\mu}{2}) \ a.s. \tag{23}$$

$1 < \check{\zeta}$, *whenever $z_0 \neq 0$, where*

$$\hat{\varsigma} = \max_{1 \leq i \leq m} \varsigma_i, \check{e} = \max_{1 \leq i \leq m} e_i, \hat{e} = \min_{1 \leq i \leq m} e_i.$$

**Proof.** By condition, we can obtain that

$$2z^T Af(x) = 2 \sum_{i,j=1}^{m} z_i b_{ij} f_j(z_j)$$

$$\leq 2 \sum_{i,j=1}^{m} |z_i| |b_{ij}| \varsigma_j |z_j| \leq \check{\zeta} \sum_{i,j=1}^{m} |b_{ij}| (z_i^2 + z_j^2)$$

$$= \check{\zeta} [\sum_{i=1}^{m} (\sum_{j=1}^{m} |b_{ij}|) z_i^2 + \sum_{j=1}^{m} (\sum_{i=1}^{m} |b_{ji}|) z_j^2]$$

$$= \check{\zeta} [\sum_{i=1}^{m} e_i z_i^2 + \sum_{j=1}^{m} e_j z_j^2] = 2\check{\zeta} z^T E z,$$

and

$$2z^T[-Ez + Bf(z)] + tr[g^T(z)Pg(z)] \leq -2(1 - \check{\zeta})z^T E z. \tag{24}$$

Therefore, in the case $1 \geq \xi$,

$$2z^T[-Ez + Bf(z)] + tr[g^T(z)Pg(z)] \leq [-2\hat{e}(1 - \xi) + \mu]|z|^2. \tag{25}$$

When $1 \geq \xi$, applying Theorem 2 with $P$ being the identity matrix, we can deduce that

$$\limsup_{k \to \infty} \frac{1}{k} ln(|z(k, z_0)|) \leq -(\rho_1 - H\rho_2 + \hat{e}(1 - \xi) - \frac{\mu}{2}) \ a.s. \tag{26}$$

When $1 < \xi$,

$$2z^T[-Ez + Bf(z)] + tr[g^T(z)Pg(z)] \leq [-2\hat{e}(1 - \xi) + \mu]|z|^2. \tag{27}$$

It follows from Theorem 2 again that

$$\limsup_{k \to \infty} \frac{1}{k} ln(|z(k, z_0)|) \leq -(\rho_1 - H\rho_2 - \check{e}(\xi - 1) - \frac{\mu}{2}) \ a.s. \tag{28}$$

We complete the proof. □

*3.2. Stabilization by Linear Uncertain Stochastic Perturbation*

We are aware that neural network

$$\dot{u}_k = -Eu_k + Bf(u_k)$$

can sometimes be unstable. It may be assumed that subjecting an unstable neural network to an uncertain stochastic perturbation would cause it to behave even worse, or become more unstable. However, this is not always the case. Uncertain stochastic perturbation can actually make an unstable neural network more stable. In this section, we will demonstrate that any neural network of the form (3) can be stabilized by uncertain stochastic perturbation. For practical purposes, we will only consider linear uncertain stochastic perturbations. This means that we will only focus on perturbations of the form:

$$g(z(k))dW_k = \sum_{l=1}^{n} G_l z(k)dW_l(k), h(z(k))dC_k = \sum_{l=1}^{n} H_l z(k)dC_l(k)$$

i.e., $g(z) = (G_1 z, G_2 z, \cdots, G_n z)$, $h(z) = (H_1 z, H_2 z, \cdots, H_n z)$, where $G_l, H_l, 1 \leq l \leq n$ are all $m \times m$ matrices. In this case, the uncertain stochastic perturbed network (8) becomes

$$\begin{cases} dz(k) = [-Ez(k) + Bf(z(k))]dk + \sum_{l=1}^{n} G_l z(k)dW_l(k) + \sum_{l=1}^{n} H_l z(k)dC_l(k), k \geq 0 \\ z(0) = z_0 \in R^m. \end{cases} \tag{29}$$

Note that

$$tr[g^T(z)Pg(z)] = \sum_{l=1}^{n} z^T G_l^T PG_l z, tr[h^T(x)Ph(z)] = \sum_{l=1}^{n} z^T H_l^T PH_l z,$$

$$z^T Pg(z)g^T(z)Pz = tr[g^T(z)Pzz^T Pg(z)]$$

$$= \sum_{l=1}^{n} z^T G_l^T Pz^T PG_l z = \sum_{l=1}^{n} (z^T PG_l z)^2,$$

and

$$z^T Ph(z)h^T(x)Pz = tr[h^T(z)Pzz^T Ph(z)]$$

$$= \sum_{l=1}^{n} z^T H_l^T P z^T P H_l z = \sum_{l=1}^{n} (z^T P H_l z)^2.$$

The proof can be obtained easily by Theorem 2, which we omit here.

**Theorem 6.** *Assume there exists a symmetric positive definite matrix $P = (p_{ij})_{m \times n}$ and some constants $\mu \in R$ and $\rho_1, \rho_2, H \geq 0$ such that*

$$2z^T[-Ez + Bf(z)] + \sum_{l=1}^{n} z^T G_l^T P G_l z \leq \mu z^T P z \tag{30}$$

*and*

$$\sum_{l=1}^{n} (z^T P G_l z)^2 \geq \rho_1 (z^T P z)^2,$$

$$\sqrt{\sum_{l=1}^{n} (z^T P H_l z)^2} \leq \frac{\rho_2}{n} z^T P z$$

*for all $z \in R^m$. Then, the solution of Equation (8) satisfies*

$$\limsup_{k \to \infty} \frac{1}{k} ln(|z(k, z_0)|) \leq -(\rho_1 - H\rho_2 - \frac{\mu}{2}) \ a.s. \tag{31}$$

*whenever $z_0 \neq 0$. Especially, if $\rho_1 - H\rho_2 > \mu/2$, then the stochastic neural network (8) is almost surely exponentially stable.*

*3.3. Some Examples*

**Example 1.** *Let*

$$G_l = \zeta_l I, H_l = \vartheta_l I, \ 1 \leq l \leq n,$$

*where $\zeta_l, \vartheta_l, 1 \leq l \leq n$ are all real numbers and $I$ is the identity matrix. Then, Equation (29) becomes*

$$dz(k) = [-Ez(k) + Bf(z(k))]dk + \sum_{l=1}^{n} \zeta_l z(k) dW_l(k) + \sum_{l=1}^{n} \vartheta_k z(k) dC_l(k). \tag{32}$$

*The parameters $\zeta_l, \vartheta_l, 1 \leq l \leq n$ denote the strength of the stochastic and uncertain perturbations, respectively. By selecting the identity matrix as the value of $P$, we observe that*

$$\sum_{l=1}^{n} z^T G_l^T P G_l z = \sum_{l=1}^{n} |G_l z|^2 = \sum_{l=1}^{n} \zeta_l^2 |z|^2 \tag{33}$$

*and*

$$\sum_{l=1}^{n} (z^T P G_l z)^2 = \sum_{l=1}^{n} (z^T \zeta_l z)^2 = \sum_{l=1}^{n} \zeta_l^2 |z|^4. \tag{34}$$

*Similarly, we have*

$$\sqrt{\sum_{l=1}^{n} (z^T P H_l z)^2} = \sqrt{\sum_{l=1}^{n} \vartheta_k^2 |z|^4} = \sqrt{\sum_{l=1}^{n} \vartheta_k^2 |z|^2}. \tag{35}$$

*Moreover, by* (2), *we have*

$$2z^T PAf(z) \leq 2|z|\|A\|\|f(z)\| \leq 2\check{\varsigma}\|A\||z|^2,$$

*where* $\check{\varsigma} = \max_{1 \leq l \leq m} \varsigma_l$ *and* $\|A\| = \sup\{|Az| : z \in R^m, |z| = 1\}$. *Hence,*

$$2z^T P[-Ez + Bf(z)] \leq 2(\check{\varsigma} - \hat{e})|z|^2, \tag{36}$$

*where* $\hat{e} = \min_{1 \leq l \leq m} e_l$. *By combining Equations* (33)–(36) *and utilizing Theorem* 6, *we can conclude that the solution to Equation* (32) *meets*

$$\limsup_{k \to \infty} \frac{1}{k} \ln(|z(k, z_0)|) \leq -(\sum_{l=1}^{n} \zeta_l^2 - nH\sqrt{\sum_{l=1}^{n} \vartheta_k^2} - (\check{\varsigma}\|A\| - \hat{e})), a.s.$$

*whenever* $z_0 \neq 0$. *Especially, if*

$$\sum_{l=1}^{n} \zeta_l^2 - nH\sqrt{\sum_{l=1}^{n} \vartheta_k^2} > \check{\varsigma}\|A\| - \hat{e}$$

*hold, then the uncertain stochastic neural network* (32) *is almost surely exponentially stable.*

**Remark 3.** *If we set* $\zeta_l = 0$ *for* $2 \leq l \leq n$, *then Equation* (32) *simplifies even further to*

$$dz(k) = [-Ez(k) + Bf(z(k))]dk + \zeta_1 z(k)dW_1(k) + \vartheta_1 z(k)dC_1(k),$$

*here, we just rely on a Wiener–Liu process scalar as the origin of the uncertain stochastic perturbation. This uncertain stochastic network is almost surely exponentially stable provided*

$$\zeta_1^2 - H\vartheta > \check{\varsigma}\|A\| - \hat{e}.$$

*The neural network described by* $\dot{u}_k = -Eu_k + Bf(u_k)$ *can be stabilized by incorporating a sufficiently strong and uncertain stochastic perturbation in a particular way. In other words, we can draw the corollary that this simple example illustrates.*

**Corollary 3.** *If* (2) *is satisfied, a Wiener–Liu process can stabilize any neural network with the given form*

$$\dot{u}_k = -Eu_k + Bf(u_k).$$

*Notably, it is also feasible to utilize a single scalar Wiener–Liu process for this purpose.*

**Example 2.** *For each l, choose a positive definite* $m \times m$ *matrix* $U_l$ *and* $V_l$ *such that*

$$z^T U_l z \geq \frac{\sqrt{3}}{2}\|U_l\||z|^2, \quad z^T V_l z \leq \frac{1}{2}\|V_l\||z|^2.$$

*There are numerous matrices that meet the criteria or characteristics being discussed. Let* $\zeta$ *be a real number and define* $G_l = \zeta U_l$. *Let* $\vartheta$ *be a real number and define* $H_l = \vartheta V_l$. *Then, Equation* (29) *becomes*

$$dz(k) = [-Ez(k) + Bf(z(k))]dk + \zeta\sum_{l=1}^{n} U_l z(k)dW_l(k) + \vartheta\sum_{l=1}^{n} V_l z(k)dC_l(k). \tag{37}$$

*And let P be the identity matrix, noting that*

$$\sum_{l=1}^{n} z^T G_k^T P G_k z = \sum_{l=1}^{n} |\zeta U_l z|^2 \leq \zeta^2 \sum_{l=1}^{n} \|U_l\|^2 |z|^2,$$

$$\sum_{l=1}^{n} (z^T P G_k z)^2 = \zeta^2 \sum_{l=1}^{n} (z^T U_l z)^2 \geq \frac{3\zeta^2}{4} \sum_{l=1}^{n} \|U_l\|^2 |z|^4$$

*and*

$$\sqrt{\sum_{l=1}^{n} (z^T P H_l z)^2} = \sqrt{\vartheta^2 \sum_{l=1}^{n} (z^T V_l z)^2} \leq \frac{\vartheta}{2} \sqrt{\sum_{l=1}^{n} \|V_l\|^2} |z|^2.$$

*By merging (36) with the above and then utilizing Theorem 6, we can deduce that the solution to (37) satisfies*

$$\limsup_{k\to\infty} \frac{1}{k} log(|z(k, z_0)|) \leq -(\frac{3\zeta^2}{4} \sum_{l=1}^{n} \|U_l\|^2 - \frac{1}{2} n H \vartheta \sqrt{\sum_{l=1}^{n} \|V_l\|^2} - (\xi \|A\| - \hat{e})) \ a.s. \quad (38)$$

*whenever $z_0 \neq 0$. So, if*

$$\frac{3\zeta^2}{4} \sum_{l=1}^{n} \|U_l\|^2 - \frac{1}{2} n H \vartheta \sqrt{\sum_{l=1}^{n} \|V_l\|^2} \geq (\xi \|A\| - \hat{e}),$$

*then the uncertain stochastic neural network (37) is almost surely exponentially stable.*

**Example 3.** *We examine the scenario where the network's dimension, denoted as m, is an even number, specifically $m = 2q(q \geq 1)$. Suppose we set n to 1, meaning we select a scalar Wiener–Liu process $(W_1(k), C_1(k))$. Additionally, let $\zeta$ be a real number and P the identity matrix again; then, we define that*

$$G_1 = \begin{pmatrix} 0 & \zeta & & & 0 \\ -\zeta & 0 & & & \\ & & \ddots & & \\ & & & 0 & \zeta \\ 0 & & & -\zeta & 0 \end{pmatrix}, H_1 = \begin{pmatrix} 0 & \vartheta & & & 0 \\ -\vartheta & 0 & & & \\ & & \ddots & & \\ & & & 0 & \vartheta \\ 0 & & & -\vartheta & 0 \end{pmatrix}.$$

*Then, Equation (29) becomes*

$$dz(k) = [-Ez(k) + Bf(z(k))]dk + \zeta \begin{pmatrix} z_2(k) \\ -z_1(k) \\ \vdots \\ z_{2q}(k) \\ -z_{2q-1}(k) \end{pmatrix} dW_1(k) + \vartheta \begin{pmatrix} z_2(k) \\ -z_1(k) \\ \vdots \\ z_{2q}(k) \\ -z_{2q-1}(k) \end{pmatrix} dC_1(k).$$

$$(39)$$

*Note that*

$$z^T G_1^T P G_1 z = \zeta^2 |z|^2, \ (z^T P G_1 z)^2 = 0 \quad (40)$$

*and*

$$2z^T P[-Ez + Bf(z)] \leq 2(\xi \|A\| - \hat{e})|z|^2. \quad (41)$$

*By integrating (40) with (41), and subsequently utilizing Theorem 6, we can derive that the solution to (39) meets:*

$$\limsup_{k\to\infty} \frac{1}{k} \ln(|z(k, z_0)|) \leq -(\frac{1}{2} \zeta_k^2 - (\xi \|A\| - \hat{e})), a.s.$$

*whenever $z_0 \neq 0$. So, the uncertain stochastic neural network* (39) *is almost surely exponentially stable if $\zeta_k^2 > 2(\xi - \hat{e}\|A\|)$.*

**Remark 4.** *Different from the almost sure exponential stability of stochastic Hopfield neural networks based on the probability theory of additive measures [6,12], uncertain stochastic Hopfield neural networks are more complex in terms of handling conditions and processes of almost sure exponential stability, such as the conditions of Theorems 2–6. In addition, we use the Itô–Liu formula, Liu inequality (Lemma 1), the Liu lemma (Lemma 2), etc, and these conclusions are all obtained using subadditive measures.*

**Remark 5.** *The practical significance of almost sure exponential stability in uncertain stochastic Hopfield neural networks is that it ensures robust and reliable performance in real-world applications, such as image or speech recognition, financial analysis, or control systems. Almost sure exponential stability enables the network to reliably handle uncertainties and variations in the input data. It improves the neural network's ability to generalize and make accurate predictions, even when faced with Liu noises and Wiener noises. This stability increases the neural network's practical usefulness and applicability in real-world scenarios.*

## 4. Conclusions

The main focus of this paper is the stability of Hopfield neural network dynamical systems with uncertain stochastic perturbations. The paper presents a theorem for judging the stability of such systems, along with two conclusions of sufficient conditions for stability. The stability of neural network systems with linear uncertain stochastic perturbations is studied in order to facilitate the discussion. We note that uncertain stochastic neural networks can be divided into two types: one is uncertain stochastic neuron activation functions, such as the Boltzman machine model, and the other is neural networks with uncertain stochastic weighted connections. Therefore, when considering uncertain stochastic neural networks, both of these cases should be considered. The uncertain stochastic neural network model studied in the paper is the second type, which involves neural networks with uncertain stochastic weighted connections. Overall, this paper provides a valuable contribution to the field of neural networks by considering the effects of both stochastic and uncertain elements on network stability and proposing methods for analyzing such systems. This work can also extend to the two-layer cellular neural network,impulsive model, or the reaction diffusion model, as in Refs [30–32]. There is currently no corresponding research result for neural networks using uncertain stochastic neuron activation functions, uncertain stochastic two-layer cellular neural network, the uncertain stochastic impulsive model, or the reaction diffusion model, and researchers can develop these areas in the near future.

**Author Contributions:** Conceptualization, Z.J. and C.L.; methodology, Z.J. and C.L.; software, Z.J. and C.L.; validation, Z.J. and C.L.; formal analysis, Z.J.; investigation, Z.J. and C.L.; writing—original draft preparation, Z.J.; writing—review and editing, Z.J. and C.L.; supervision, C.L. All authors have read and agreed to the published version of the manuscript.

**Funding:** This work was supported in part by the Natural Science Foundation of Ningxia (no. 2020AAC03242), Major Projects of North Minzu University (no. ZDZX201805), Governance and Social Management Research Center of Northwestic regions, and Nation and First-Class Disciplines Foundation of Ningxia (Grant No. NXYLXK2017B09).

**Data Availability Statement:** Not applicable.

**Conflicts of Interest:** The authors declare no conflict of interest.

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
