# Peer review of "Almost Sure Exponential Stability of Uncertain Stochastic Hopfield Neural Networks Based on Subadditive Measures"

_mathematics, doi:10.3390/math11143110_

Round 1

Reviewer 1 Report

see attachment.

fine

Author Response

Dear Reviewer

Detailed modification instructions can be found in the attachment.

Reviewer 2 Report

The paper includes major results in the field. The subject is appropriate for the journal and what is reported is new and timely with the current literature. The introduction is well presented and the paper does appear in a good journal style.  The previous suggestion given to the authors allow to improve the paper and to make it more interesting for more readers. In fact it is useful to remark pracical application about the reported result in the field both of applied science or engineering. Moreover globally the paper merit after a revision to be published. In my opinion the analytical part is well developed and without errors.

The paper sounds interesting and the results could be furtherly applied in distributed networks by usinn neurons.

I suggest to strongly evaluate the possibility in using the proposed approach to CNN base networks where the neuron model is very easy.

Therefore also to increase the reader interest I suggest to include the following paper:

IEEE Transactions on Circuits and Systems I: Fundamental Theory and ApplicationsVolume 45, Issue 2, Pages 157 - 1621998 Document type Article Source type Journal ISSN 10577122 DOI 10.1109/81.661681 View more   

Self-organization in a two-layer CNN

  • Arena, Paolo;
  • Baglio, Salvatore;
  • Fortuna, Luigi;
  • Manganaro, Gabriele

Save all to author list   I think that in a further revised version the paper xcould be of hig intesest.

Author Response

(The authors gave the same response as above.)

Reviewer 3 Report

    1.The abstract does not clearly explain the proposed procedure. Abstract needs to be revised. Not clear at all on what work had been done in the paper, what results been achieved, significance of results achieved when compared to existing literature. 2. Please check twice formatting errors and sentence formation/grammar errors. 3. Please describe in detail with existing literature, why results achieved in paper are better than previous work or studies. 4. Please add few lines to illustrate the importance of results and compare with literature in the conclusion.

5.Authors should add table of momenclature so that readers could relate to symbols used in the paper easily.

Author Response

(The authors gave the same response as above.)

Reviewer 4 Report

The authors of this paper studied the almost sure exponentially stability of uncertain stochastic Hopfield neural networks based on subadditive measure of chance theory. The content of the paper is interesting, and the obtained results are valuable. However, there are some comments that should be considered during the revision of the manuscript:

Comments:

1.    The authors could provide an appropriate reference for Definition 7 used in the paper.

2.    What is the necessity of the usage of the axioms 1-4. Give a note on it.

3.    It could be better if the authors add more discussion in the numerical examples.

4.    The motivation and highlights of the paper should be strengthened further.

5.    The linguistic quality of the paper should be enhanced further.

6.    The authors could add a remark to discuss the practical significance of the paper.

Language needs further improvement.

Author Response

(The authors gave the same response as above.)

Round 2

Reviewer 4 Report

The paper can be accepted.